# Peritrophin-like Genes Are Associated with Delousing Drug Response and Sensitivity in the Sea Louse *Caligus rogercresseyi*

**DOI:** 10.3390/ijms232113341

**Published:** 2022-11-01

**Authors:** Antonio Casuso, Gustavo Núñez-Acuña, Valentina Valenzuela-Muñoz, Constanza Sáez-Vera, Cristian Gallardo-Escárate

**Affiliations:** 1Interdisciplinary Center for Aquaculture Research (INCAR), Universidad de Concepción, Concepción 4030000, Chile; 2Laboratory of Biotechnology and Aquatic Genomics, Department of Oceanography, Universidad de Concepción, Concepción 4030000, Chile

**Keywords:** *Caligus rogercresseyi*, peritrophin, delousing drugs, transcriptome, ectoparasite

## Abstract

*Caligus rogercresseyi* is the main ectoparasite that affects the salmon industry in Chile. The mechanisms used by the parasite to support its life strategy are of great interest for developing control strategies. Due to the critical role of insect peritrophins in host–parasite interactions and response to pest control drugs, this study aimed to identify and characterize the peritrophin-like genes present in *C. rogercresseyi*. Moreover, the expression of peritrophin-like genes was evaluated on parasites exposed to delousing drugs such as pyrethroids and azamethiphos. Peritrophin genes were identified by homology analysis among the sea louse transcriptome database and arthropods peritrophin-protein database obtained from GenBank and UniProt. Moreover, the gene loci in the parasite genome were located. Furthermore, peritrophin gene expression levels were evaluated by RNA-Seq analysis in sea louse developmental stages and sea lice exposed to delousing drugs deltamethrin, cypermethrin, and azamethiphos. Seven putative peritrophin-like genes were identified in *C. rogercresseyi* with high homology with other crustacean peritrophins. Differences in the presence of signal peptides, the number of chitin-binding domains, and the position of conserved cysteines were found. In addition, seven peritrophin-like gene sequences were identified in the *C. rogercresseyi* genome. Gene expression analysis revealed a stage-dependent expression profile. Notably, differential regulation of peritrophin genes in resistant and susceptible populations to delousing drugs was found. These data are the first report and characterization of peritrophin genes in the sea louse *C. rogercresseyi*, representing valuable knowledge to understand sea louse biology. Moreover, this study provides evidence for a deeper understanding of the molecular basis of *C. rogercresseyi* response to delousing drugs.

## 1. Introduction

The copepod ectoparasite *Caligus rogercresseyi*, also known as sea louse, represents one of the major problems in the Chilean salmon aquaculture [1,2]. This copepod feeds on fish mucus and blood [3], producing skin lesions, weight loss, and immunosuppression in fish [4,5]. Salmon farming economic costs generated by this ectoparasite have motivated research on the mechanisms behind *C. rogercresseyi* infection on salmonid fishes. A significant step forward was the publication of the transcriptome of the parasite development stages [6] and the publication of the sea louse genome draft [7]. Functional genomics data have provided information on genes involved in the interaction between this parasite and the host fishes [8,9]. Moreover, sea lice control has been historically based on delousing drugs, highlighting the interest in understanding drug response mechanisms [10,11]. However, the genetic mechanisms involved in these processes in *C. rogercresseyi* have not yet been fully described.

Peritrophins are proteins with a history of participating in parasitic mechanisms and as part of the response to insecticide drugs [12,13]. In insects, peritrophins have an important role in host–parasite interaction [14,15]. For example, an increased expression of peritrophins was found as part of the host innate immune response after infections by the scabies mite [16]. Additionally, an upregulation of peritrophin genes was detected in the mosquitoes *Lutzomyia longipalpis* artificially fed with blood containing *Leishmania* parasite [17]. On the other hand, peritrophin proteins participate in the protective response against toxic compounds damaging [18,19,20]. For instance, the peritrophin AeIMUC1 removes the heme groups from the host species to avoid the deleterious effects of free radical formation in the African malaria mosquito *Anopheles gambiae* [12,21]. Moreover, upregulation of the peritrophin-1-like gene was observed in the potato pest *Leptinotarsa decemlineata* exposed to the biological insecticide Spinosad [13]. Regarding sea lice species, a transcriptomic study conducted in *Lepeophtheirus salmonis* exposed to the chitin synthesis inhibitor Lufenuron showed a downregulation of transcripts containing chitin-binding domains in the affected parasites, including genes with the peritrophin-A domain that is characteristic of the structural domains of peritrophic membrane proteins [22].

The peritrophins are chitin-binding proteins originally isolated from insect intestinal peritrophic membrane or peritrophic matrix (PM) [23,24]. The PM is a non-cellular structure covering the midgut of various arthropods [25,26,27], including copepod parasitic species *Lernaea cyprinacea*, *Ergasilus orientalis,* and *Neoergasilus japonicus* [28]. The PM is similar to a sieve with physiological functions that include intestine epithelium protection against mechanical damage, neutralization of ingested toxins or pathogens, and compartmentalization of the digestive process [29,30,31,32,33,34]. The PM structure is composed of chitin fibrils associated with a protein matrix [18,35]. Among the structural proteins of this matrix, peritrophins are the most abundant, comprising 35 to 55% of the structure [18,36,37,38]. In addition, the peritrophins have functional domains which are tightly associated with the chitin fibrils of the PM [39,40]. The most recurrent domains reported are the chitin-binding domains with N-glycosylations sites and mucin domains with O-glycosylation sites [41]. In crustaceans, the presence of PM and homologous insect peritrophin genes have been described [23,42,43,44,45,46]. For instance, peritrophin-like genes have been identified in crustaceans such as shrimp [23,44,45,47,48,49] and Chinese mitten crab [50,51]. Moreover, Lai and Aboobaker [24] reported a new family of 80 peritrophin-like genes from a transcriptomic database of 55 crustaceans from the Malacostraca class.

The lack of information regarding the *C. rogercresseyi* peritrophins leaves relevant information gaps for controlling the pathogen. Currently, the main sea louse control methods are based on delousing drugs such as pyrethroids, organophosphates, and chitin inhibitors [52,53]. However, a reduction in the efficacy of these compounds has already developed in *C. rogercresseyi* populations [54,55,56,57,58]. Thus, several studies have investigated the underlying mechanisms involved in sea lice response to delousing drugs, suggesting potential resistance mechanisms [10,11,59,60,61,62]. Nevertheless, the importance of the PM and the role of peritrophins in the response of *C. rogercresseyi* to delousing drugs have not been described yet. Given the importance of peritrophin proteins in host–parasite interaction and their role in providing a physical barrier against toxic compounds, this study aimed to identify and characterize the peritrophin-like genes present in *C. rogercresseyi* developmental stages, and we additionally evaluated the modulation of peritrophin-like genes in *C. rogercresseyi* exposed to cypermethrin, deltamethrin, and azamethiphos.

## 2. Results

### 2.1. Identification and Characterization of Putative Peritrophin-likes Genes

From in silico analysis of the *C. rogercresseyi* transcriptome, 16 contigs were identified with homology for peritrophin genes (Appendix A). Seven of these sequences presented the full open reading frame (ORF). The identified sequences were consecutively named Cr_Peritrophin1 (*Cr_Per1*) to Cr_Pereritrophin7 (*Cr_Per7*). The longest ORF was *Cr_Per4* with 1008 bp, while the shortest ORF was Cr_Per2 with 315 bp (Table 1). *C. rogercresseyi* peritrophin sequences show high identity with the other peritrophin-like proteins reported in crustaceans (Figure 1). Most clades are supported with high bootstrapping values, with ranges between 60 and 100%. The Cr_Per6 and Cr_Per2 were grouped in the same clade of *Penaeus merguiensis*_SOP and *Penaeus monodon*_SOP. Cr_Per5 was clustered together with *Palaemon carinicauda*_Per, separated from a copepods clade. The Cr_Per1 and Cr_Per3 were clustered together with *L. salmonis_*Per1 and *Eurytemora affinis*_POE-like1, respectively. A second clade contained Cr_Per7 and Cr_Per4 grouped with *Caligus clemensi_*PT.

From the characterization of peritrophin protein sequences, differences in the number of ChtBD2 domains and distribution of the conserved Cys were observed (Figure 2A). For instance, three ChtBD2 were found in Cr_Per1, Cr_Per3, Cr_Per4, and Cr_Per7, and two of these domains in Cr_Per5 and Cr_Per6 (Figure 2B). Cr_Per2 did not show any of these domains despite their high homology with peritrophin-like protein sequences. Furthermore, all the Cr_Per sequences exhibited O-glycosylation sites, while N-glycosylation sites were only found in Cr_Per3, Cr_Per5, Cr_Per6, and Cr_Per7. Protein subcellular prediction determined by protein sequences that Cr_Per1 was a cell membrane protein, and the rest of Cr_Per were soluble extracellular proteins (Table 1). Finally, the predicted tertiary structure for Cr_Per proteins, the invertebrate chitin-binding protein (d1dqca), was used as a template with a confidence of more than 96% for every Cr_Per (Appendix A).

### 2.2. Identification of Cr_Per Sequences in the Genome

The sequences of the peritrophin-like genes of *C. rogercresseyi* were located in the sea louse genome (BioProyect Accession: PRJNA551027). The characterized peritrophins were located in different chromosomes, and most of them presented a single exon (Table 2). Although, *Cr_Per5*, located in chromosome 11, presented two exons separated by a short intron of 61 pb (Appendix A). 

### 2.3. Peritrophin Expression Analysis in C. rogercresseyi Developmental Stages

The expression profile of peritrophin-like genes characterized in *C. rogercresseyi* was evaluated in all developmental stages. The RNA-Seq analysis showed two clusters of peritrophin-like genes. The first one was composed of *Cr_Per1, Cr_Per3,* and *Cr_Per5*, which were highly expressed in free live stages, and chalimus I-II. Moreover, a second cluster composed of *Cr_Per2, Cr_Per4, Cr_Per6,* and *Cr_Per7* was mainly expressed in chalimus III-IV and adults. Notably, the *Cr_Per7* was strongly upregulated in females (Figure 3A). The RT-qPCR validation was performed for *Cr_Per1, Cr_Per2, Cr_Per3*, and *Cr_Per6* (Figure 3B). *Cr_Per1* showed high expression levels in early stages (nauplius and copepodid), which were similar to the in silico analysis (Figure 3B). Furthermore, the expression levels of *Cr_Per2* and *Cr_Per6* in adults sea louse were upregulated (Figure 3B), while the *Cr_Per3* was upregulated significantly in chalimus stages (Figure 3B).

### 2.4. Peritrophin Expression in Drug-Exposed C. rogercresseyi 

Sea lice challenged with delousing chemicals showed a drug-dependent peritrophin expression profile (Figure 4). Sea lice groups exposed to pyrethroids integrated one sole cluster, upregulating *Cr_Per2, Cr_Per4*, and *Cr_Per7*. In contrast, the azamethiphos group strongly upregulated *Cr_Per1, Cr_Per3,* and *Cr_Per5* (Figure 4). Furthermore, expression was validated in susceptible and resistant *C. rogercressyi* populations to these drugs [11]. Interestingly, all drug-exposed groups evidence modulation of peritrophin genes expression (Figure 5A). In the azamethiphos group, the resistant population showed the greatest differences with *Cr_Per1*, *Cr_Per2,* and *Cr_Per3*, while the deltamethrin susceptible group modulated five of the seven *Cr_Per* genes. Interestingly, *Cr_Per7* gene expression was only upregulated in resistant sea lice populations challenged with cypermethrin (Figure 5A). Moreover, cypermethrin *Cr_Per7* was the gene with the highest fold change between populations of susceptible and resistant sea lice (Figure 5B).

## 3. Discussion

*Caligus rogercresseyi* is the most important ectoparasite in Chilean salmon farming; thereby, interest in developing new control methods has encouraged the study of their physiology and molecular mechanisms of interaction with the host. Peritrophins, as the main component of the peritrophic membrane, have been evaluated as a pharmacological target for insect control methods [63,64]. Furthermore, peritrophin proteins have been related in some insects to the protective response against damage caused by toxic compounds, such as insecticides [14,65]. Here, we identified and characterized *C. rogercresseyi* peritrophin-like genes with a potential role in *C. rogercresseyi* response to delousing drugs. 

Studies performed in arthropods have described different numbers of peritrophin gene isoforms expressed in one species. For instance, Lai and Aboobaker [24] reported 80 peritrophin-like transcripts from five orders of Malacostraca. In insect species, such as *Spodoptera frugiperda* and *Abracris flavolineata,* 38 and 23 perithrophins-like, respectively, have been reported [66]. In this study, 16 peritrophin-like sequences were identified from *C. rogercresseyi* transcriptome database [6]. The phylogenetic analysis showed divergences among *C. rogercresseyi* peritrophin-like proteins, suggesting different functions. For instance, *Cr_Per2* and *Cr_Per6* converged with peritrophin-like proteins expressed during shrimp oogenesis called Shrimp Ovarian Peritrophin (SOP) [23,45,47,67]. SOPs are the main protein in *Penaeus semisulcatus* oocytes and an important component for the development of spawned eggs [45]. Furthermore, SOPs play an immune role in protecting spawned eggs against pathogens [23]. The high expression levels of *Cr_Per2* and *Cr_Per6* in *C. rogercresseyi* females suggest an SOP-like function of these two sea louse peritrophins. Regarding *Cr_Per3*, this gene was clustered with the protein obstructor-E-like (obst-E) of the copepod *E. affinis*. Obstructing genes represent the invertebrate multigene family [68] mainly studied in insects [69,70,71]. These proteins with chitin-binding domains participate in the development of the cuticle [72]. For instance, in *Drosophila melanogaster* Obst-E participates in the contraction and expansion of the cuticle in the pre-adult stages, regulating the chitin disposition in the cuticle in the third larvae stage and participates in determining the shape of the pupa [73,74]. In *C. rogercresseyi*, a total of 479 transcripts were previously reported as proteins linked to cuticle formation, with high levels of transcription during the larval stage and highly regulated in the adults stage, and also with differential expression levels after delousing drug exposure [10]. Here, *Cr_Per3* gene was highly expressed in early sea louse stages, suggesting a function in the molting process.

Differences among *C. rogercresseyi* peritrophin isoforms were observed. For instance, the theoretical molecular weight found for the seven peritrophin-like protein sequences was between 12,141 and 38,324 KDa. These results are in concordance with previous results in crustacean species where several peritrophins sizes have been reported [23,44,51]. The Chinese mitten crab *Eriocheir sinensis* has the Es-peritrophin1 whit a MW of 28.93 KDa. However, for this crab was also reported the Es-peritrophin2 with 51.17 KDa [50]. Additionally, the pI found in the peritrophin-like proteins of *C. rogercresseyi* were in a range between 4.5 and 6, which is common among other peritrophins [39]. The extreme environment in these parts of the gastrointestinal tract requires special characteristics, and these values of pIs have been reported in other PM-related proteins [75,76]. 

The chitin-binding domains are the functional peritrophin domains reported in the PM of most invertebrates [72,77]. Herein, six peritrophin-like proteins identified in *C. rogercresseyi* presented the chitin-binding domain (ChBD2). This domain allows the assembly of the proteins and chitin fibers forming the PM [18,50,78]. Differences in the number of Cys were reported in a family of crustacean peritrophin-like proteins [24]. The consensus sequence of *C. rogercresseyi* peritrophin ChBD2 identified in this work was CX_1-23_CX_5_CX_9-10_CX_12-15_CX_7-12_C. However, *Cr_Per6* ChBD2 presented one Cys more than the other sea louse peritrophins. Furthermore, characterized *C. rogercresseyi* peritrophins presented two or three ChBD2s. Similar results have been reported in crustacean species, *Fenneropenaeus Chinensis*, *P. monodon*, *Penaeus semisulcatus,* and *E. sinensis* [44,50,51]. In addition, more than three ChBD2 domains have been described for insect peritrophins [77]. Interestingly, no presence of functional ChBD2 domains was found in the Cr_Per2 sequence. These results are consistent with those reported in *A. gambiae*, where PM proteins without ChBD2 were identified, suggesting a function associated with protein–protein interactions and formation of three-dimensional PM structure [12]. Additionally, the presence of signal peptide in the ChBD2 protein is a characteristic of chitin-binding functions in extracellular matrices [72]. The analysis of signal peptide and cellular localization of *C. rogercresseti* peritrophins suggest that they are expressed and secreted in the extracellular matrix, except for the Cr_Per1, which was determined as a membrane protein. These data are in accordance with the PM reported structure, which is a non-cellular structure composed of secreted proteins embedded in a proteoglycan matrix with chitin [79]. Interestingly, chitin synthesis inhibitors have been used for caligidosis control [80]. However, research has shown a decrease in the effectiveness of these treatments [53]. Notably, the structural characteristics of *C. rogercresseyi* peritrophins suggest their participation as part of drug response [81].

To identify the genome position of Cr_Per genes, the sequence was mapped against *C. rogercresseyi* genome (BioProject Accession: PRJNA551027). Each characterized sea louse peritrophins was located in a different chromosome. Furthermore, six of them presented a single exon. These results are in agreement with a study conducted on the common cutworm, *Spodoptera litura*, where the peritrophin-37 gene with a single exon was found in the cutworm genome [82]. Additionally, peritrophin genes such as *PMP9* of *Tribolium castaneum* [72], and the *Peritrophin-44* gene of the ectoparasite *Lucilia cuprina* [37,83] had only a single exon. However, an *L. cuprina* peritrophin called *peritrophin-95* has been described with a small intron of 77 bp [84]. In our work, a small intron of 61 bp was identified for *Cr_Per5* gene, located in sea louse chromosome 11. 

As we described above, peritrophins are essential molecules to the physiological functions of PM and play an important role in the control of toxicity damage [85,86]. Indeed, an in vitro study reported that *Aedes aegypti* peritrophin AelMUCI can bind to the hemotoxic groups [14], suggesting a detoxification mechanism used by the mosquito to reduce the damage caused by free radicals during hemoglobin digestion obtained from blood-feeding [12]. Therefore, studies aimed at the control of pests have targeted the PM in different insect species [87,88,89]. For example, increased mortality was observed when the termite *Reticulitermes flavipes* treated with double-stranded RNA (dsRNA) to reduce the expression levels of PM genes such as peritrophin and chitin, was challenged with the themicide imidacloprid [65,90]. Moreover, the transcriptome analysis of the potato pest *L. decemlineata* showed a strong upregulation of *peritrophin-1-like* gene when the beetle was exposed to the insecticide Spinosad [13]. Here, our data show a differential expression pattern of peritrophin-like genes in *C. rogercresseyi* exposed to delousing drugs azamethiphos and pyrethroids. Moreover, we present the first peritrophin genes expression analysis in populations of *C***.**
*rogercresseyi* differing in sensitivity to deltamethrin, cypermethrin, and azamethiphos. Interestingly, some genes were differentially expressed specifically after drug exposure. The expression of *Cr_Per7* gene significantly increased the expression in exposed individuals from the cypermethrin-resistant population. Similarly, an increased expression of the *Cr_Per3* gene was observed in exposed individuals belonging to the azamethiphos-resistant population. Thus, these proteins might have a function in the detoxification process, preventing the accumulation of these compounds in resistant individuals. Previously, Chávez-Mardones, Valenzuela-Muños, and Gallardo-Escárate [10] reported a high correlation between azamethiphos exposure and cuticle precursor gene expression in *C. rogercresseyi*, suggesting an influence of this organophosphate on the regulation of cuticle-related proteins in sea lice. Notably, *Cr_Per3* is structurally close to cuticle-related proteins. Another interesting finding was the upregulation of the *Cr_Per1, Cr_Per2*, *Cr_Per3,* and *Cr_Per4* genes in control groups, which suggests these genes as markers for drug sensitivity testing.

Our results support the hypothesis of the potential functionality of these proteins in decreased mortality after drug exposure. As we identified peritrophin genes that are constituents of the PM, we provide evidence for a deeper understanding of the molecular basis of *C. rogercresseyi* response to delousing drugs. Nevertheless, complementary studies will be needed to determine the role played by these peritrophin genes in the delousing drugs response. Finally, the recognition of molecular markers to predict drug resistance in sea louse can be an important asset for the salmon industry [61]. This work identified significant modulation of the *Cr_Per7* gene in exposed individuals from the resistant cypermethrin population in comparison with the susceptible population. Similarly, the *Cr_Per3* gene showed significant differences between susceptible and resistant populations to azamethiphos. Thus, the characterization of sea louse peritrophin genes are reported here, evidenced their utility in *C. rogercresseyi* control. Therefore, we suggest that these genes should be considered candidate genes for sea louse control in further studies.

## 4. Materials and Methods

### 4.1. Identification of Peritrophin-like Genes in C. rogercresseyi

#### 4.1.1. Identification and Characterization of *C. rogercresseyi* Peritrophins

From the transcriptome database described for *C. rogercresseyi* [6], peritrophin cDNA sequences were identified. Sea louse transcripts annotation was performed by tBLASTx analysis against peritrophin transcripts in CLC Genomics Workbench software (Version 22, CLC Qiagen Bioinformatics, CA, USA), using the EST database available for arthropods in GenBank. Transcripts with an E-value ≤ 1 × 10^−5^ were selected. The cDNA open reading frame (ORF) was determined and translated using Geneious software (Version 11.0.9, Biomatters Ltd., Auckland, NI, New Zealand). Moreover, a multiple alignment analysis using a BLOSUM62 matrix was performed with peritrophin sequences described for arthropods in GenBank. Furthermore, a phylogenetic tree was generated with the Neighbor-Joining method, using Jukes–Cantor as a genetic distance model, and applying a bootstrap of 1000. In addition, protein functional domains were identified with the online tools SMART [91] and PROSITE [92]. SignalP-5.0 was used to determine the presence of signal peptides [93] and DeepLoc-1.0 for predicting the protein subcellular localization [94]. The molecular weight (MW) and theoretical isoelectric point (pI) of the putative protein sequences were determined with Compute pI/Mw software [95]. Additionally, O- and N-linked glycosylation sites were predicted with NetOGlyc 4.0 Server [96] and NetNGlyc 1.0 Server [97], respectively. The Phyre2 web portal was used for modeling, prediction, and analysis of the tertiary protein structure [98]. 

#### 4.1.2. Peritrophins Genome Organization

The *C. rogercresseyi* Genome draft (Genome NCBI Accession: PRJNA551027) was used to predict the chromosomal location and genome structure of the Cr_Per genes. Re-sequencing analysis was performed using the Map Reads to Reference (1.6) plugin included in CLC Genomics Workbench (Version 22, CLC Bio, Denmark). The parameters used were mismatch cost = 2, cost of insertions and deletions = linear gap cost, insertion cost = 1, deletion cost = 1, length fraction = 0.9, and similarity fraction = 0.8.

### 4.2. Expression Analysis of Peritrophin Genes

#### 4.2.1. Ontogeny Expression 

Raw reads obtained for *C. rogercresseyi* developmental stages [6] were mapped against peritrophin contigs with a significant annotation (E-value > 10^−5^). The RNA-seq settings were a minimum length fraction = 0.8 and a minimum similarity fraction (long reads) = 0.8. The expression value was set as a Transcripts Per Million of Reads (TPM). The distance metric was calculated with the Euclidean distance method [99], where the mean expression level in 5–6 rounds of k-means clustering was subtracted. Finally, multi-factorial statistics analysis based on a negative binomial GLM was used to compare gene expression. 

#### 4.2.2. Sea Lice Challenged with Delousing Drugs

Peritrophin contigs were used as reference to map the raw reads obtained from *C. rogercresseyi* populations exposed to drugs: 8 ppb of azamethiphos (Byelice^®^, Bayer Cono Sur, Santiago, Chile), 3 ppb of deltamethrin (AMX^®^, Pharmaq South America, Santiago, Chile), and 5 ppb of cypermethrin (Betamax^®^, Novartis Chile S.A., Santiago, Chile) [11]. RNA-seq analysis parameters and statistical tests were performed as described in Section 4.2.1.

#### 4.2.3. RT-qPCR Validation

Peritrophin genes expression in ontogeny and drug exposure were validated by RT-qPCR. For ontogeny, *C. rogercresseyi* samples were obtained from the experimental laboratory of the Marine Biological Station, University of Concepción, Dichato, Chile. Sea lice larvae were kept in trays provided with a flow of seawater at 12 °C and soft aeration. Atlantic salmons were infected to obtain juvenile and adult stages at a load of 35 copepodites per fish. Samples of each stage of the parasite were collected: nauplius, copepodite, chalimus, adult females, and males. Furthermore, peritrophin genes expression was validated using *C. rogercresseyi* strains previously characterized as susceptible and resistant to each delousing drug: deltamethrin, cypermethrin, and azamethiphos [11]. For each delousing drug, an exposed plus a control group was chosen for the susceptible and the resistant population. Each group consisted of 30 individuals (fifteen females and fifteen males). The exposure period for azamethiphos was 30 min (100 ppb), 40 min for deltamethrin (3 ppb), and 30 min for cypermethrin (15 ppb). Once the trial time was over, the parasites were fixed in the RNALater solution (Ambion^®^, Thermo Fisher Scientific^TM^, Waltham, MA, USA) and stored at −80 °C.

Total RNA was isolated using Trizol reagent (Ambion, Life Technologies™, Carlsbad, CA, USA) according to the manufacturer’s instructions and quantified on Nanodrop One spectrophotometer (Thermo Scientific, Waltham, MA, USA). The RT-qPCR standardization was carried out according to the MIQE guidelines [100]. cDNA was synthesized starting at 200 ng/µL of initial total RNA and using the RevertAid H Minus First Strand cDNA Synthesis kit (Thermo Fisher Scientific, MA, USA), following the manufacturer’s directions. The RT-qPCR reaction was performed on the StepOnePlus™ thermocycler (Applied Biosystems, Foster City, CA, USA). For the expression quantification of peritrophin-like genes, the comparative ΔΔCt method was used [101]. The data were normalized using the β-tubulin II gene as housekeeping [8]. The PowerUp™ SYBR™ Green Master Mix (Thermo Fisher Scientific, Waltham, MA, USA) was used in a final reaction volume of 10 uL. Amplification was carried out under the following conditions: 95 °C for 10 min, 40 cycles of 95 °C for 15 s, 30 s at 60 °C, followed by a melting curve (95 °C for 15 s, 60 °C for 1 min, and 95 °C for 15 s). Specific primers were designed in Geneious (Version 11.0.9, Biomatters Ltd., Auckland, NI, New Zealand) (Table 3). Statistical analyses were performed in the GraphPad Prism 9.0 software (San Diego, CA, USA). The data present the mean of the standard error (SEM). Significant differences were determined by one-way ANOVA and Tukey post-hoc analysis. Statistically significant values were set at *p*-values < 0.05.

## 5. Conclusions

This is the first study to identify and characterize peritrophin-like genes in the sea louse *C. rogercresseyi*, and the expression profile of these genes in response to delousing drugs. *C. rogercresseyi* peritrophin-like genes were differentially expressed in drug-susceptible and drug-resistant sea lice populations exposed to delousing drugs azamethiphos, deltamethrin, and cypermethrin. The characterized sea louse peritrophins encode extracellular proteins with putative functions as a structural component of the peritrophic membrane or cuticula. Gene expression is differentially regulated during the parasite developmental stages, suggesting different functions through ontogeny. Further studies will be conducted to evaluate the use of peritrophin genes as a tool for sea louse control.

## Figures and Tables

**Figure 1 ijms-23-13341-f001:**
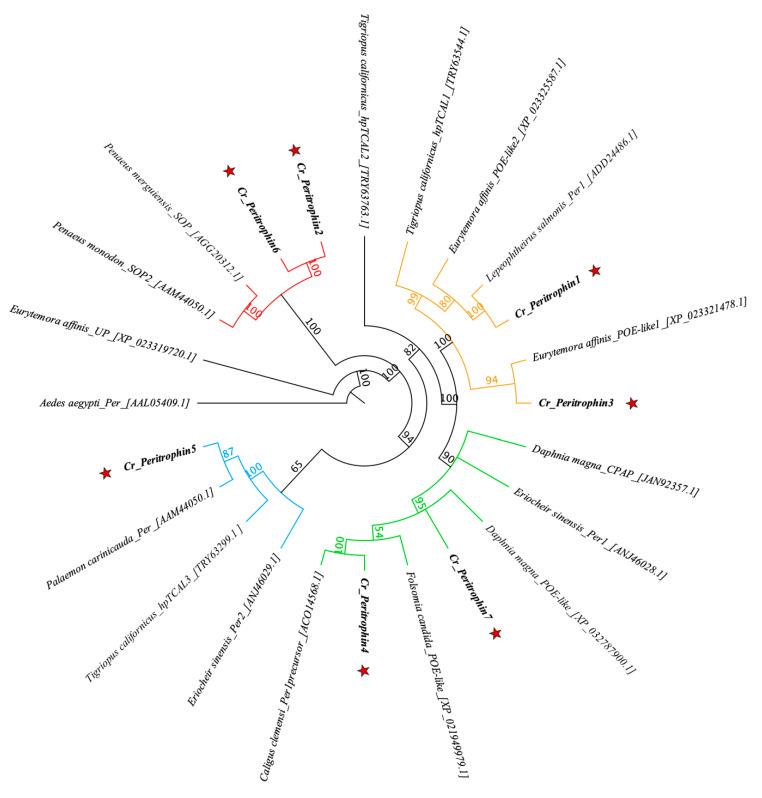
Phylogenetic analysis of sea louse peritrophin proteins. Jukes–Cantor genetic distance model was used to determine the bootstrapping indicated in the nodes. Sequence access numbers are found next to species names. The Cr_Per sequences are marked with a red star. The different clades are shown in red, orange, green, and blue.

**Figure 2 ijms-23-13341-f002:**
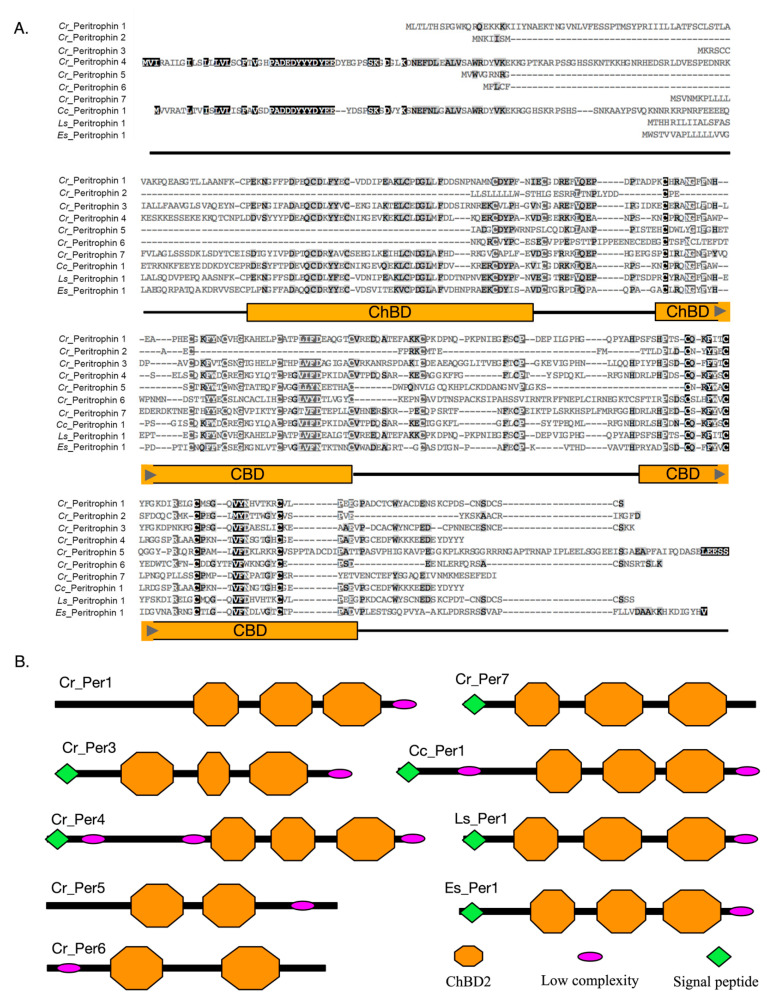
Amino acid sequences analysis of peritrophins identified in *C. rogercresseyi*. (**A**)**.** Multiple alignments of the Cr_Per protein with other crustacea peritrophin proteins deposited in GenBank. The dark gray region indicates the positions where all the sequences share the same amino acid residue, and oranges boxes represent chitin-binding domains. (**B**)**.** The predicted domain structure of the peritrophin-like proteins of *C. rogercresseyi* (Cr). Cc: *Caligus clemensi* [ACO14568.1], Ls: *Lepeophtheirus salmonis* [ADD24486.1], Es: *Eriocheir sinensis* [ANJ46028.1].

**Figure 3 ijms-23-13341-f003:**
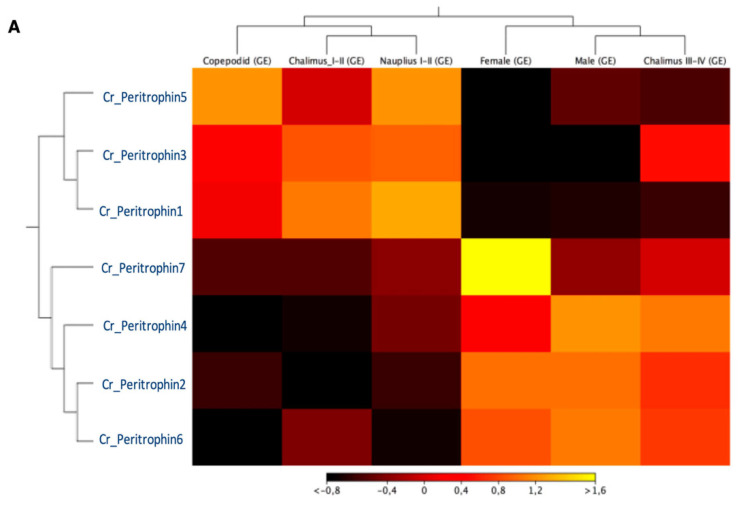
Transcriptional expression of contigs annotated as peritrophin genes during *C. rogercresseyi* developmental stages. (**A**)**.** Heatmap showing the TPM values using transcriptomic data of *C. rogercresseyi* developmental stages. Hierarchical clustering was based on Euclidean distances with an average linkage. (**B**)**.** RT-qPCR validation of four *Cr_Per* in sea louse developmental stages. Bars represent the relative expression abundance of the transcripts (mean ± standard deviation). The lowercase letters above the error bars for each gene showed significant differences at *p* < 0.05.

**Figure 4 ijms-23-13341-f004:**
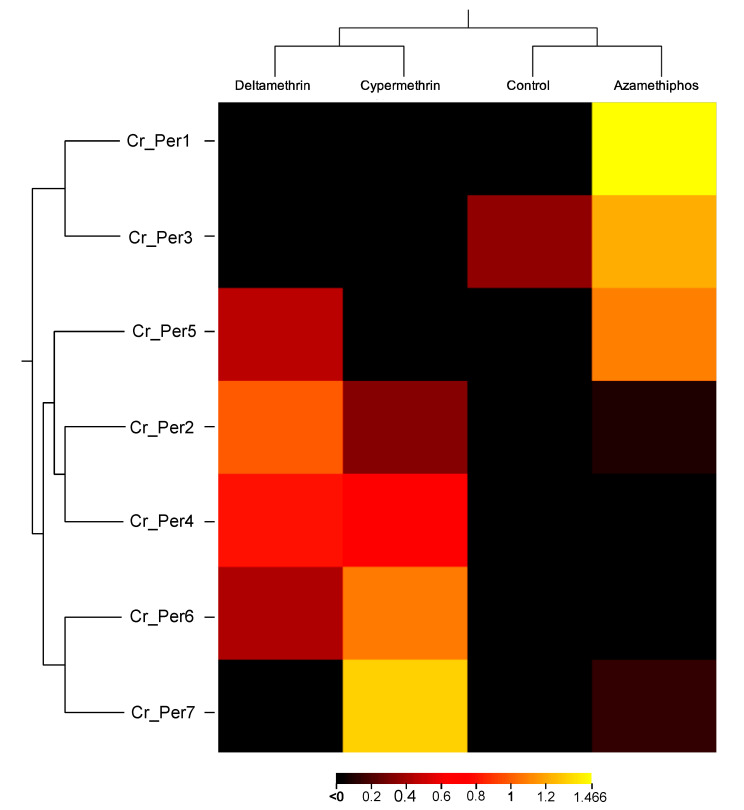
Heatmap of gene transcription values from drugs-exposed *C. rogercresseyi* to 100 ppb of azamethiphos, 3 ppb deltamethrin, and 15 ppb cypermethrin. Transcript abundance is represented as TPM values. Color scales show relative transcript expression. Hierarchical clustering was conducted based on Euclidean distance for TPM data with an average linkage.

**Figure 5 ijms-23-13341-f005:**
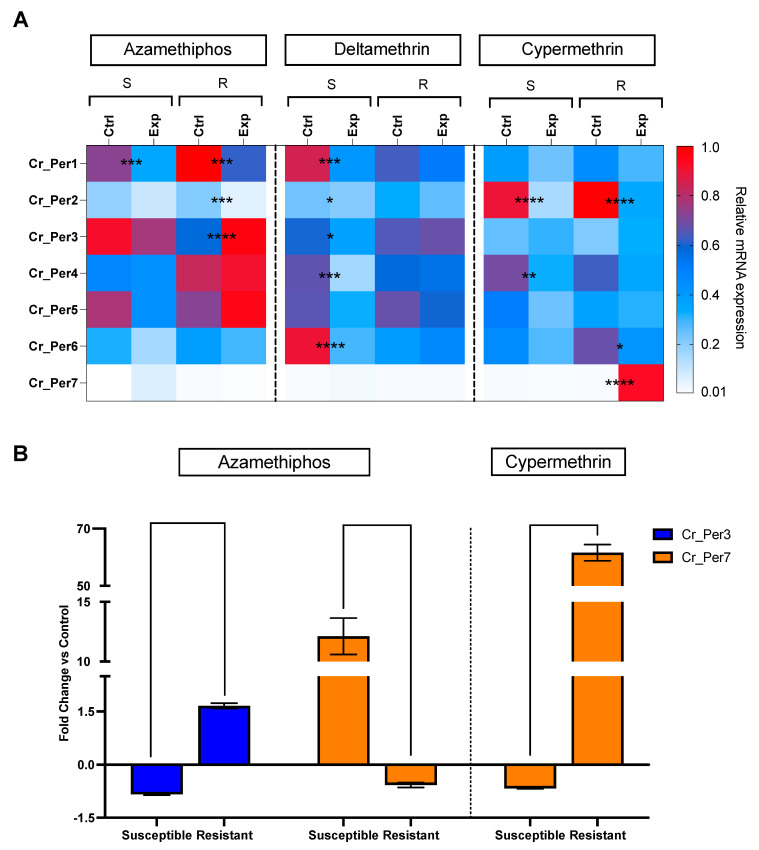
RT-qPCR of peritrophin genes in *C. rogercresseyi* populations susceptible (S) and resistant (R) to drugs. (**A**). Heatmaps show the relative expression (fold change values) of Cr_Per in control (Ctrl) and exposed (Exp) lice to azamethiphos, deltamethrin, and cypermethrin. The relative expression levels were normalized with β tubulin as an endogenous control. Asterisks indicate statistically significant values between control and exposed lice of the same population and treatment, where *p*-value < 0.05 (*), < 0.005 (**), < 0.0003 (***), and < 0.0001 (****). Color scales show relative gene expression. (**B**). Peritrophin genes with higher fold change in *C. rogercresseyi* exposed to drugs. Column bars represent the means for the fold change differences based on the relative expression of the exposed group against the control group for each population. Data are presented as the mean expression ± standard error (SEM). Asterisks indicate statistical differences between the resistance and susceptible control group for each population.

**Table 1 ijms-23-13341-t001:** *C. rogercresseyi* peritrophins characterization.

*C. rogercresseyi* Peritrophins	ORF Length (bp)	Translate Sequence (aa)	MW (KDa)	pI	α Helix (%)	β Strand (%)	O-Linked Glyc	N-Linked Glyc	Protein Subcellular Prediction
Cr_Per1	924	307	34.163	5.08	9	24	6	Not found	Cell membrane
Cr_Per2	315	104	12.141	5.04	37	21	1	Not found	Extracellular
Cr_Per3	768	255	27.881	4.65	11	20	3	1, *	Extracellular
Cr_Per4	1008	335	38.324	5.4	3	30	13	*	Extracellular
Cr_Per5	663	220	24.117	5.97	1	20	9	1, *	Extracellular
Cr_Per6	591	196	22.352	4.57	14	34	3	1	Extracellular
Cr_Per7	783	260	29.61	5.33	10	16	7	1	Extracellular

ChtBD2: chitin-binding domains; *: low probability site.

**Table 2 ijms-23-13341-t002:** Genome structure of *C. rogercresseyi* peritrophin-like.

*C. rogercresseyi* Peritrophins	Gene Copies	Chromosome	No. of Exons	Start Position (nt)	Final Position (nt)
Cr_Per1	1	11	1	32,016,274	32,016,768
Cr_Per2	1	3	1	26,035,334	26,035,644
Cr_Per3	1	19	1	11,957,508	11,958,268
Cr_Per4	1	12	1	14,772,262	14,772,731
Cr_Per5	1	11	2	32,986,846	32,987,474
Cr_Per6	1	9	1	5,738,200	5,738,500
Cr_Per7	1	6	1	5,486,856	5,487,637

**Table 3 ijms-23-13341-t003:** Oligonucleotide primers used for RT-qPCR analysis.

Primer Name	Sequence	Mold Sequence	Tm (°C)	Efficiency (%)
Cr_Peritrophin1_F	TCTTCAACCATGAGGCACCC	contig10	60	112.4
Cr_Peritrophin1_R	GGCCTCGTCGAAGATCAGAG			
Cr_Peritrophin2_F	CCCGAAGACTCACCAATCCC	contig 4784	60	118.6
Cr_Peritrophin2_R	GGCAGTCGAAGGAACACTCA			
Cr_Peritrophin3_F	TCGTGAGTTTGTCCAGGAGC	contig 6390	60	112.2
Cr_Peritrophin3_R	AAGGAAGCTCATGTCCCGTG			
Cr_Peritrophin4_F	GGACTATGAGGGCCCTTCCT	contig 12860	60	110
Cr_Peritrophin4_R	GGTGGGGCCTTTCTTTTCCT			
Cr_Peritrophin5_F	CCCCTTGGAAAGTCCTGCAA	contig0002090	60	105
Cr_Peritrophin5_R	CGCTTGCGGAGTTTGTCAAA			
Cr_Peritrophin6_F	GCAAATCAATCCCAGCCCAC	contig4172	60	112.6
Cr_Peritrophin6_R	CCGAGGGGCGTATCGTAAAA			
Cr_Peritrophin7_F	AGATCCAAACGCCCAGAGTG	contig8519	60	109
Cr_Peritrophin7_R	GGAGCAGTCTTCTGGGTGAC			

## Data Availability

Not applicable.

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
