# Peer review of "Peritrophin-like Genes Are Associated with Delousing Drug Response and Sensitivity in the Sea Louse *Caligus rogercresseyi"

_ijms, 2022, doi:10.3390/ijms232113341_

Round 1
Reviewer 1 Report
The manuscript identifies and characterizes C. rogercresseyi peritrophins. Furthermore, it reveals significant modulation of the Cr_Per7 gene in exposed individuals from the resistant cypermethrin population if compared to the susceptible. Similarly, different expression of the Cr_Per3 gene was found in susceptible and resistant populations to azamethiphos. The work provides a basis for further investigation of the role of peritrophins in resistance to antiparasitic agents. The manuscript is well written and can be accepted for publication.
Minor editorial remarks:
Line 51: “the scabies mite infections [16]” – please, delete “infection”. There is a repetition.
Lines 51-53: Please, revise the sentence, it is not enough clear.
Line 210: “gene isoforms expressed in one specie” – “specie” should be “species”
Author Response
Dear reviewer,
We do appreciate your suggestions to improve our manuscript. Please find below the responses to every question and/or comment that you have made. Furthermore, the corrections and suggested changes to the manuscript were made using “Track Changes” to facilitate its visualization. Also, the manuscript is highlighted in different colors for each reviewer’s comment.
Yours Sincerely,
The corresponding author
Reviewer 1 (green highlighted)
Minor editorial remarks:
Line 51: “the scabies mite infections [16]” – please, delete “infection”. There is a repetition. R. Thanks for the comment, the sentence was corrected (line 51).
Lines 51-53: Please, revise the sentence, it is not enough clear. R. Thanks for the comment, the sentence has been written again (lines 52 - 54).
Line 210: “gene isoforms expressed in one specie” – “specie” should be “species” R. Thanks for the comment, the sentence was corrected (line 209).
Reviewer 2 Report
The manuscript is well written and well structured with clear layout. It can be accepted for publication almost as is, pending very few minor corrections, which are outlined below.
L35 - "C. rogercressey" should be given in full name at first mention in the introduction
L52 - "feeding with blood-fed Leishmania-infected": I am not sure I am catching the meaning of this sentence. Do authors mean mosquitoes fed with Leishmania-infected blood? If yes, slightly rephrase to improve clarity; otherwise, can You explain please?
L89 - Check the use of "Peritrophin" or "Peritrophins" throughout the manuscript. The latter is OK when used alone, whilst it should be the former when preceding another name. e.g. "perithrophins proteins" should be replaced by "Perithrophin proteins" and so on
L210 - species instead of specie
L211 - insect species instead of insects species, Also, check for similar misspellings thoroughout the manuscript
L251 - Crustacean species, there is no need to use the saxon genitive here, Check for similar issues throughout the manuscript
L252-253 - Can authors explain more clearly what this statement means?
L266-267 - "...their participation as part of this nature drug response": Please, Slightly rephrase to improve clarity
L309-310 - In response to what? Please, be more specific
L316 - "...provided evidenced of their utility...": Please, simplify as "evidenced their utility"
L317-318 - Candidate genes for what? Please, explain what do You mean
L409-410 - Overall, the senetnce is sufficiently clear, but can authors sliglthly rephrase it to further improve its clarity?
Figure 5A: The scalebar right to the figure is completely white and thus readers can only guess what values are high or low
Author Response
Dear reviewer,
We do appreciate your suggestions to improve our manuscript. Please find below the responses to every question and/or comment that you have made. Furthermore, the corrections and suggested changes to the manuscript were made using “Track Changes” to facilitate its visualization. Also, the manuscript is highlighted in different colors for each reviewer’s comment.
Yours Sincerely,
The corresponding author
Reviewer 2 (blue highlighted)
The manuscript is well written and well structured with clear layout. It can be accepted for publication almost as is, pending very few minor corrections, which are outlined below.
L35 - "C. rogercressey" should be given in full name at first mention in the introduction R. Thanks for the comment, full name was included in the sentence as you suggested (line 35).
L52 - "feeding with blood-fed Leishmania-infected": I am not sure I am catching the meaning of this sentence. Do authors mean mosquitoes fed with Leishmania-infected blood? If yes, slightly rephrase to improve clarity; otherwise, can You explain please? R. Thanks for the comment, we were referring to mosquitoes fed on blood infested with Leishmania. The sentence was rephrased for better understanding, as you suggested (lines 52 - 54).
L89 - Check the use of "Peritrophin" or "Peritrophins" throughout the manuscript. The latter is OK when used alone, whilst it should be the former when preceding another name. e.g. "perithrophins proteins" should be replaced by "Perithrophin proteins" and so on R. Thanks for the comment. The writing throughout the manuscript was corrected (line 77).
L210 - species instead of specie R. Thanks for the comment, the sentence was corrected.
L211 - insect species instead of insects species, Also, check for similar misspellings thoroughout the manuscript. R. Thanks for the comment. The writing throughout the manuscript was corrected.
L251 - Crustacean species, there is no need to use the saxon genitive here, Check for similar issues throughout the manuscript R. Thanks for the comment. Following your suggestion, the saxon genitive was corrected throughout the manuscript.
L252-253 - Can authors explain more clearly what this statement means? R. Thanks for the question. We wanted to point out that more than three ChBD2 domains have been described in some insect peritrophins. The sentence was corrected (line 268-269).
L266-267 - "...their participation as part of this nature drug response": Please, Slightly rephrase to improve clarity. R. Thanks for the comment, the sentence was corrected (292 -294).
L309-310 - In response to what? Please, be more specific R. Thanks for the question. We refer to the response to delousing drugs. The sentence was corrected, as you suggested (341 - 343).
L316 - "...provided evidenced of their utility...": Please, simplify as "evidenced their utility" R. Thanks for the comment, the sentence was corrected (line 348).
L317-318 - Candidate genes for what? Please, explain what do You mean R. Thanks for the comment, the sentence was corrected to improve clarity (348 -350).
L409-410 - Overall, the senetnce is sufficiently clear, but can authors sliglthly rephrase it to further improve its clarity? R. Thanks for the comment. The sentence was rephrase, as you suggested (446 -448).
Figure 5A: The scalebar right to the figure is completely white and thus readers can only guess what values are high or low R. Thanks for the observation. The highest and lowest value were included in the scalebar of Figure 5A, as you suggested.